# Beliefs, attitudes and practices towards scabies in central Ghana

Yaw Ampem Amoako[1,2,3]*, Lotte Suzanne van Rietschoten[3], Michael Ntiamoah Oppong[1], Kwabena Oppong Amoako[1], Kabiru Mohammed Abass[4], Bernard Akoto Anim[5], Dennis Odai Laryea[6], Richard Odame Phillips[1,2], Ymkje Stienstra[3,7]

**1** Kumasi Centre for Collaborative Research in Tropical Medicine, Kwame Nkrumah University of Science and Technology, Kumasi, Ghana, **2** School of Medicine and Dentistry, Kwame Nkrumah University of Science and Technology, Kumasi, Ghana, **3** Department of Internal Medicine/Infectious Diseases, University of Groningen, University Medical Center Groningen, Groningen, The Netherlands, **4** Agogo Presbyterian Hospital, Agogo, Ghana, **5** Kokofu Government Hospital, Kokofu, Ghana, **6** Disease Surveillance Department, Ghana Health Service, Accra, Ghana, **7** Department of Clinical Sciences, Liverpool School of Tropical Medicine, Liverpool, United Kingdom

* yamoako2002@yahoo.co.uk

## Abstract

### Background

Scabies commonly affects poor populations in low-middle-income countries. The WHO has advocated for country-driven and country-owned control strategies. Knowledge of context specific issues will be important for design and implementation of scabies control interventions. We aimed to assess beliefs, attitudes and practices towards scabies in central Ghana.

### Methodology/Principal findings

Data was collected via semi-structured questionnaires for people who had active scabies or scabies in the past year and people who never had scabies in the past. The questionnaire covered several domains: knowledge about the causes and risk factors; perceptions towards stigmatisation and consequences of scabies in daily life; and treatment practices. Out of 128 participants, 67 were in the (former) scabies group and had a mean age of 32.3 ± 15.6 years. Overall scabies group participants less often indicated a factor to predispose to scabies than community controls; only 'family/friends contacts' was more often mentioned in the scabies group. Scabies causation was attributed to poor hygiene, traditional beliefs, heredity and drinking water. Individuals with scabies delay care seeking (median time from symptom onset to visiting the health centre was 21 [14 – 30] days) and this delay is enhanced by their beliefs (like witchcraft and curses) and a perception of limited disease severity. Compared to past scabies participants in the dermatology clinic, participants with past scabies in the community tended to have a longer delay (median [IQR] 30 [14–48.8] vs 14 [9.5–30] days, p = 0.002). Scabies was associated with health consequences, stigma, and loss of productivity.

### Conclusion/Significance

Early diagnosis and effective treatment of scabies can lead to persons less frequently associating scabies with witchcraft and/ or curses. There is the need to enhance health education

**Data Availability Statement:** All relevant data have been included in the manuscript.

**Funding:** The authors received no specific funding for this work.

**Competing interests:** The authors declare they have no competing interests.

to promote early care seeking, enhance knowledge of communities on impact and dispel negative perceptions about scabies in Ghana.

## Author summary

Scabies, a skin Neglected Tropical Disease (NTD) characteristically causes an intensely pruritic rash. Scabies commonly affects individuals living in poor, overcrowded communities with limited access to healthcare. Integrated control of skin NTDs requires an understanding of local context issues. We explored understanding, beliefs and care seeking behaviours towards scabies in central Ghana.

We found that individuals with scabies delay care-seeking and this delay is enhanced by their beliefs and a perception of limited disease severity. Compared to past scabies participants in the dermatology clinic, participants with past scabies in the community tended to have a longer delay. Participants in the scabies group recruited from community were more likely to associate the disease with curses, witchcraft, heredity or drinking water, than those recruited from the dermatology clinic. Scabies was associated with health consequences, stigma, and loss of productivity.

Early diagnosis and effective treatment of scabies can lead to persons less frequently associating scabies with supernatural forces and promote care-seeking in the formal health system. Health education to promote awareness and early care-seeking is essential to dispel negative perceptions about scabies in Ghana. Further, health systems should be capacitated to adequately treat affected individuals and their contacts to facilitate scabies control.

## Introduction

Scabies, an intensely pruritic skin disease, is caused by the mite *Sarcoptes scabiei* and presents an enormous burden in resource limited settings [1]. It is common among persons living in poor and overcrowded communities but is also a common condition among patients presenting to hospital with dermatologic conditions in resource poor settings [2, 3]. Scabies has also been reported among prisoners, preschool children [4] and within communities of Africa [5–7]. The greatest burden of scabies is in low- and middle-income countries where overcrowding and inadequate access to effective treatment serve as drivers of disease transmission. On-going scabies transmission within communities is related to delayed treatment, as a result of delayed diagnosis, poor access to treatment, non-adherence with treatment or a lack of a systematic approach to treating household contacts [5, 8].

There are challenges to scabies control in rural Africa. First, there are few expert dermatologists. Further, there is under-recognition and under-diagnosis among health staff due in part to an absence of a clearly defined criteria for diagnosis [1, 2, 7, 9] and the disease predominantly affects persons of low socioeconomic status living in rural communities with limited access to healthcare [5]. Added to these may be beliefs, attitudes and health seeking behaviours that may adversely impact scabies control by perpetuating stigma and delaying initiation of appropriate treatment.

In some endemic communities, awareness about scabies is low. Itching is a common symptom of scabies [5, 6]; however, in a previous study from Pakistan, half of the participants did not know that itching may be due to scabies although 67.5% knew that scabies was curable and preventable. Furthermore, the majority of the patients suffering from scabies did not know the mode and duration of transmission of scabies, its prevention, and treatment [10]. A lack of

awareness can be associated with delay in diagnosing scabies and health seeking among affected persons and this can further potentiate on-going community spread. Poverty and stigmatisation of affected persons can result in delays in seeking care and treatment [11]. In a study of patients accessing a health centre in Brazil, only 51.8% of scabies patients sought medical assistance on account of their condition. Awareness among health practitioners was also poor as the skin disease was only diagnosed when it was pointed out by the patient to the healthcare practitioner [12].

The WHO has designated scabies as a Neglected Tropical Disease (NTD) and encouraged control measures. Mass Drug Administration (MDA) using ivermectin as a control tool among endemic population has revealed promising results [13–15]. The support of endemic communities is critical for control and eradication activities within affected populations. Integration of psychosocial support services have been advocated as essential in the control of neglected tropical diseases [16–19]. An understanding of the beliefs and attitudes of community members affected with scabies will be essential to guide the design and implementation of control activities. This study therefore aimed to examine health seeking behaviours towards scabies among (former) scabies patients and community controls in Ghana.

## Methods

### Ethics statement

Prior to all interviews the research was verbally explained to the participants. Written informed consent was obtained from all participants and in case of participants below 18 years old, consent was also obtained from a parent or legal guardian. Ethical approval for the study was obtained from the Committee on Human Research and Publication Ethics of the School of Medicine and Dentistry of the Kwame Nkrumah University of Science and Technology (approval number: CHRPE/AP/671/19) in Ghana and the University Medical Center Groningen Institutional Review Board (approval number 201900650) in the Netherlands. The study was conducted in accordance with the ethical principles on research involving human subjects as set out in the Declaration of Helsinki [20].

### Study settings

Participants were recruited from two locations in the Ashanti region of Ghana: communities in Asante Akim North district and from the dermatology clinic at Kokofu Government Hospital located in the Bekwai Municipal area.

The Asante Akim North district is one of the forty-three districts in Ashanti region and has a population of 68,186 according to the 2010 Ghana Population and Housing Census. About 53.5 percent of the population is rural. Males constitute 48.8 percent and females represent 51.2 percent. The district has relatively youthful population with slightly more than half (50.6%) of the population below 20 years. The average household size in the district is 4.5 persons per household. Of the population 11 years and above, 79.2% are literate. Majority of the employed population, (61%) are engaged in agriculture (mainly farming and fishing) while 16.8% are service and sales workers. The Agogo Presbyterian Hospital is the only secondary health facility in the district which receives referrals from health centres and other primary level facilities within the district [21].

### Study design and questionnaire

We conducted a cross-sectional study of knowledge, beliefs and attitudes regarding scabies in Ghana from September to November 2020. In this study, data was collected via semi-

structured, questionnaire-based interviews with participants in the selected locations. Data were collected separately for people who had active scabies or scabies in the past year and people who never had scabies in the past. Two questionnaires, one for community members who had scabies at the time of the interview or in the past year (scabies group) and one for community members who never had scabies before (non-scabies group), were developed based on existing studies on community perceptions and help-seeking behaviour on skin NTDs in West Africa [22–24]. The questionnaire covered several domains: demographics and access to health care; knowledge about the causes and risk factors; perceptions towards stigmatisation and consequences of scabies in daily life; and perceptions on self-treatment, treatment practices from traditional healers and health care centres and help-seeking behaviour.

The knowledge domain consisted of an open question on the cause of scabies and followed by closed questions whereby participants were asked whether they thought different risk factors (e.g. a parasite, hygiene, the environment, witchcraft/sorcery, curse) played a role in getting scabies. Stigmatisation was expressed in whether participants with scabies felt treated or looked upon differently or, in the non-scabies group, whether they expected this to happen in their communities; this was asked as a closed (yes/no) question.

Prior to conducting the study, all study staff received training on administering the questionnaire, with training provided by the lead member of the study team (YAA). We tested the data collection tool on 6 persons in the outpatient clinic of the Kokofu Government Hospital to assess the comprehension, acceptability and relevance of the items in the tool; no major modifications were deemed necessary after the pilot.

## Participant recruitment

We intended to include around 100 participants ((past) scabies and controls) to get a first impression of the beliefs and attitudes towards scabies in Ghana. The sample size was intended to both generate information on geographical variation and differences between scabies patients and controls. Furthermore, data saturation in the open questions was included in the decision making on the sample size.

After arrival in a community, the study team asked for the Community based surveillance volunteer (CBSV) at a public gathering point and the aim of the research was explained. Thereafter, participants were recruited through walking a random route around the village by a study team member and the CBSV. People they met coincidently were asked face-to-face to take part in the study. Any person, with or without scabies in the past year could be included in this study. For participants with active scabies, the diagnosis was made based on a history of pruritus and the typical appearance and distribution of the rash and/ or the presence of burrows. For participants with past scabies, the diagnosis was based on a self-reported history of clinically evident disease in the preceding 1 year. Participants were recruited until there was almost an equal number of individuals with scabies and controls to meet targeted recruitment number. In the hospital, participants were randomly contacted at the waiting room; the study was introduced to them after which they were invited to participate. Participants from the community were asked if they had any active rash following which the rash was examined by one of the trained members of the team to diagnose scabies. All clinical examinations were conducted in private (in the clinic or community centre as appropriate) to ensure confidentiality. Questionnaires were administered in local languages and the responses were recorded using a REDCap based form which was hosted in a database located at the University Medical Center Groningen, Netherlands.

Participants with active scabies at recruitment were treated with benzyl benzoate as per the national guidelines [25]. Briefly, this involves application of benzyl benzoate over the whole

body (except the face) twice and left overnight on two consecutive nights. The first application is done after a bath with the application repeated the next day (without a bath) and washed off 24 hours later.

## Data analysis

Data in REDCap was exported to SPSS for Windows version 20 statistical software for analysis. Descriptive statistics were used to summarize the characteristics of the study participants. Categorical variables were summarized using frequencies and percentages. In addition, the degree of association was evaluated using chi-square ($\chi2$) or Fisher's exact tests, where appropriate, with a p value of $\leq 0.05$ deemed to be statistically significant.

## Results

### Characteristics of participants

One hundred and twenty eight (128) individuals were included in the study. There were 67 participants in the scabies group. Of this, 37 were recruited from a dermatology clinic and 30 were from the community setting. There were 61 participants in the non-scabies group: 28 were from clinic setting and 33 were from the community/ non-clinic setting. The mean age (±SD) of participants was 32.3 ± 15.6 years and 30.1 ± 11.0 years for participants in the scabies and non-scabies groups respectively. There were 48.4% (62/128) females. Thirty percent of the participants had tertiary education, 14.1% had no education and 22.7% were farmers.

### Clinical features of disease (scabies cases)

Seventeen (17) individuals had scabies at the time of recruitment and 50 had scabies in the past year [median (IQR) 45 (27–90) days ago]. The complaints in patients currently having scabies and participants with scabies episode in the past year were similar, with the main clinical signs being itching (98.5%) and rash (92.5%). The median (interquartile range, IQR) duration for participants to find out their complaints were due to scabies was 21 (7.8–30) days. In 41 out 67 participants, the diagnosis of scabies was suggested by a nurse or doctor. Less frequently (9 out of 67) the diagnosis was suggested by a family member (3), teacher (2), community health volunteer (3) or community member (1). The median (IQR) time from onset of scabies complaints till a visit to health centre to seek care was 21 (14–30) days.

### Perceived aetiological factors of scabies

Overall scabies group participants less often indicated a factor to predispose to scabies than community controls; only 'family/friends contacts' was more often mentioned in the scabies group (Fig 1). According to respondents, poor hygiene practices may contribute to scabies causation. Further, having close contact with infected persons or not taking regular baths were behaviours linked to scabies. Persons who worked in children wards of hospitals or in unclean environments were also considered at risk of scabies.

Within the scabies group, the disease was mentioned to be due to a parasite (13.4% vs 23.9%) or a small organism (14.9% vs 20.9%) and friends/ family you have (23.9% vs 26.9%) respectively for participants recruited from the dermatology clinic and community settings (Fig 2).

Traditional beliefs including witchcraft and curses were more often mentioned by community controls as contributing to scabies causation. Among participants in the scabies group, 10.5% and 11.9% believed the disease could be caused by curses and witchcraft respectively compared to 27.9% and 20% in the non-scabies group (p = 0.003). Participants in the scabies

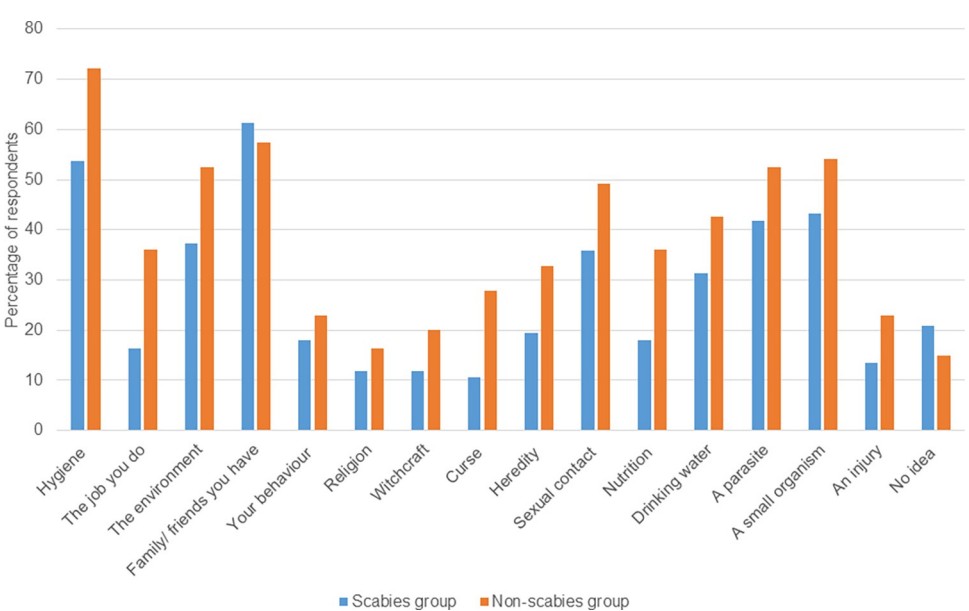

**Fig 1. Factors causing scabies according to participants (scabies group versus non-scabies group).**

group recruited from community/ non-clinic setting were more likely to associate the disease with curses (10.4% versus 0%, p = 0.002), witchcraft (11.9% versus 0%, p = 0.001), heredity (16.4% versus 3.0%, p = 0.001) or drinking water (20.9% versus 17.5%, p = 0.001), than those recruited from the dermatology clinic (Fig 2). These findings were confirmed by the data obtained in the open questions:

"witches can transmit the disease to you during sleep" (*29 years, female, clinic setting, scabies group*)

"someone can curse you with scabies" (*37 years, male, control, community setting*)

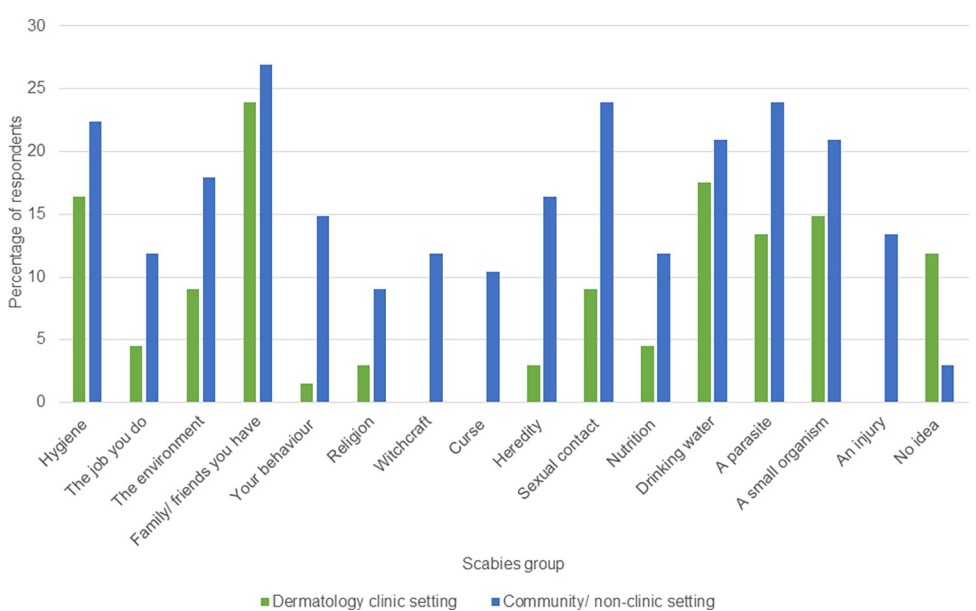

**Fig 2. Factors causing scabies in participants in the dermatology clinic versus in the community.**

Other risk behaviours described were having close or intimate contact with infected persons, and playing in overcrowded environments:

"intimate body contact during sex can lead to spread of scabies" (*28 years, female, community setting, scabies group*)

"Engaging in unprotected sex with someone who has scabies can result in scabies infection" (*34 years, male, community setting, non-scabies group*)

## Perceptions: Health consequences and stigma

Participants with scabies reported intense itching that was usually worse at night and interfered with their sleep. Additional consequences mentioned were social restriction and loss of productivity.

". . .itching especially at night. My friends don't come near me as much as they used to do before I had scabies" (*23 years, male, with scabies recruited from community/ non-clinic setting*)

"It makes me not sleep at night, so it affects my productivity at work" (*26 years, male, clinic setting*)

". . .I can not have sexual intercourse with my husband" (*49 years, female, currently with scabies, clinic setting*)

Among the scabies group, 16.4% (11/67) reported experiencing stigma because of their condition. These were generally perceived as feelings of rejection, but one participant reported teasing/ shaming.

"People did not want to come close to me; even close friends and family shunned my company" (*23 years, male, with scabies in the past year, community/ non clinic setting*)

"People did not want to talk to me. They were avoiding me and not sharing their books with me" (*16 years, female, clinic setting*)

"My friends teased me greatly and avoided contact with me" (*30 years, male, clinic setting*)
Practices:

## Health seeking behaviour

There was a delay in care seeking among participants in the scabies group. The median (IQR) time from onset of scabies complaints till a visit to health centre to seek care was 21 (14–30) days. Compared to past scabies participants in the dermatology clinic, participants with scabies in the past year in the community tended to have a longer delay in care seeking after the onset of scabies complaints (median [IQR] 30 [14–48.8] vs 14 [9.5–30] days, p = 0.002). The delay was due to several factors including a perception that the condition was not severe enough to warrant a visit to the doctor for care and the cost of treatment. Other options that were mentioned from the open questions on delayed care seeking included pregnancy (a 24 year old woman with scabies, community setting), lack of awareness about cause of disease (a 33 year old female and 29 year old man with scabies, community setting), expired health insurance (a 32 year old man, with scabies, community setting), a busy work schedule (a 28 year old female, with scabies, community setting) and absence of health facilities within community (42 year old male, community setting) (Fig 3).

One scabies patient had travelled far from the Western region to seek care in Ashanti region. Most respondents in the scabies group (53/67 or 79.1%) preferred to attend the health centre to see a health care worker rather than a traditional healer for their scabies. Reasons for this preference (Fig 4) included the opinion of others (30/67) and a confidence in services provided at the health centre (42/67). Participants in the scabies group recruited from the dermatology clinic were more likely to seek care from the formal health system due to a confidence in the services provided (73.0% versus 50.0%, p = 0.001), possession of health insurance

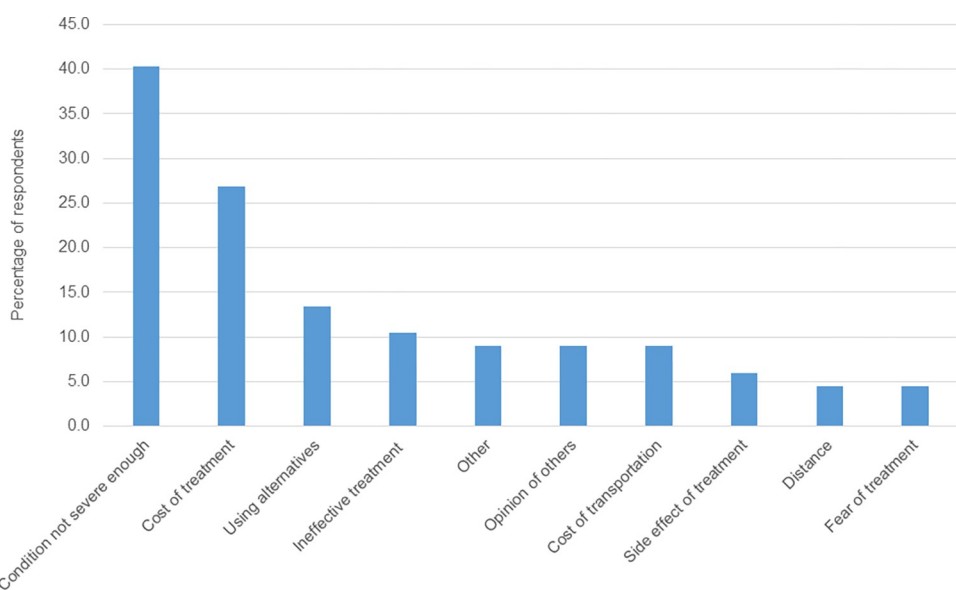

**Fig 3. Reasons for delayed care seeking among scabies group (community setting and dermatology clinic).** *Other reasons for delayed care seeking included pregnancy (1), lack of awareness about cause of disease (2), expired health insurance (1), busy work schedule (1) and absence of health facilities in community.

(40.5% versus 13.3%, p = 0.001), positive opinion of others (56.8% versus 30.0%, p = 0.001) than those recruited from the community. Additional reasons for seeking formal healthcare included positive experiences of previous patients and failed earlier treatments. Only 6/67 considered proximity to health centre as an important factor when making decisions. Only one person indicated the traditional healer as the preferred health care provider. The one person who visited a traditional health spent 30 Ghana cedis (~$3). Patients who visited a doctor or nurse spent a median of 20 Ghana cedis on the treatment (IQR 17.5–50). Data on the cost of transportation to seek care was not collected.

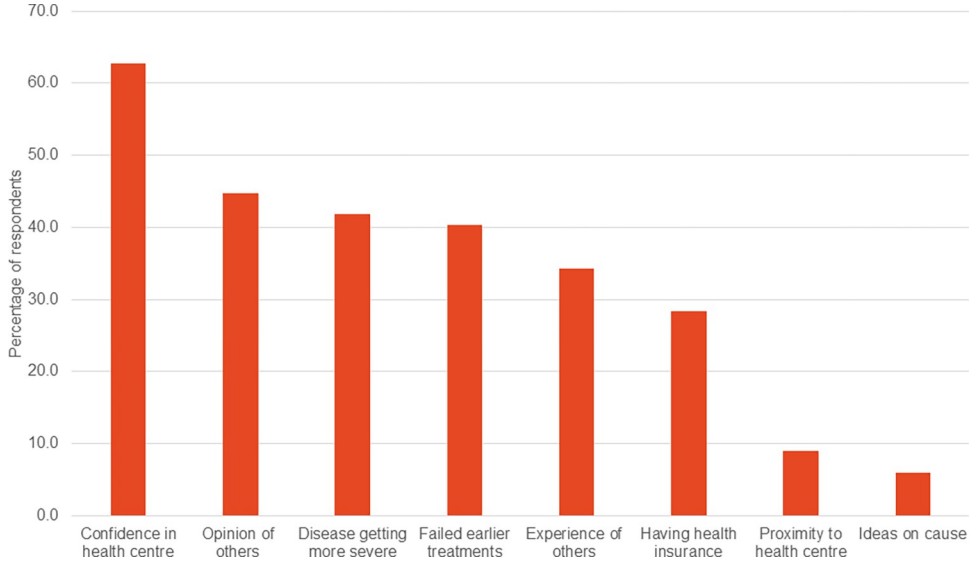

**Fig 4. Facilitators of care seeking in the formal health system by participants in the scabies group.**

Participants in the non-scabies group indicated they would seek care from the health centre if they had scabies but others mentioned that they would seek care from the local pharmacy (drug store) or engage in self-treatment:

"I would go to the hospital if I had such severe disease. I trust them to care for me" (*27 years, female, community member*)

### Practices

**Treatment of scabies.** All seventeen individuals who had scabies at recruitment were treated with benzyl benzoate as per national guidelines. Participants (59.5% or 22/37) with scabies in the past year recruited from the dermatology clinic reported receiving topical permethrin for treatment and 2 received oral ivermectin. A minority of (17.9% or 12/67) participants reported use of herbal medication and other local ointments (tree barks, *chocho cream* and "*Maame Dagomba*") to relieve symptoms at onset of disease; 8 of the persons in this category were in the group recruited from the community setting.

Participants (14/61) without scabies mentioned they would engage in self-treatment if they had scabies. Practices mentioned for self-treatment included visiting the local pharmacy to purchase ointments, taking injections, herbal medication (drinking concoction brewed from leaves or barks of trees), use of herbal creams, use of anti-itch medications and cleaning rashes with water.

"I will buy ointment from a street vendor and apply to the rashes" (*47 years, male, community member in the non-scabies group*)

"I would take some injections and use antiseptic to wash my clothes" (*23 years, male, community member in the non-scabies group*)

## Discussion

We found that individuals with scabies delay care-seeking and this delay is enhanced by their beliefs and a perception of limited disease severity. Compared to past scabies participants in the dermatology clinic, participants with past scabies in the community tended to have a longer delay. Participants in the scabies group recruited from community were more likely to associate the disease with curses, witchcraft, heredity or drinking water, than those recruited from the dermatology clinic.

Scabies symptoms, primarily itching appear approximately four weeks from the time of contact [26]. A person with scabies is considered infectious if treatment has not been initiated. In this study, the median time taken to seek care after onset of scabies symptoms was 21 days. This observed delay in health seeking has implications for ongoing community transmission with the potential to contributing to possible outbreaks which will overburden the already stretched health systems in low resource settings like Ghana.

An understanding of the beliefs, attitudes and practices on scabies is vital to the successful development and implementation of control programmes for scabies but there is a glaring dearth of data on the subject in Ghanaian context. We have explored perceptions about causation, health seeking practices, stigma and impact of scabies in central Ghana. Studies on other neglected tropical diseases of the skin in West Africa [24, 27, 28] have reported perceptions that skin NTDs are caused by lack of personal and environmental hygiene and similar findings were found in this study. More participants in the non-scabies group than the scabies group attributed causation to lack of hygiene. An appreciable proportion of participants reported scabies to be caused by a parasite, microorganism, or "friends/family you have". Scabies was also believed by some respondents to be caused by supernatural forces like witchcraft or curses. A higher proportion of participants in the non-scabies group mentioned the role of supernatural

forces. This finding can hamper care seeking and scabies control efforts in endemic communities. There is the need to enhance community education on scabies and its risk factors to dispel such misconceptions about the disease.

Furthermore, within the scabies group, there was a difference in the perception of the role of supernatural forces in disease causation; participants recruited from the community tended to associate the disease with witchcraft (p = 0.001), and curses (p = 0.002) than those recruited in the clinic setting. This difference in belief systems may explain why participants with past scabies in the community had a longer delay in care seeking following onset of scabies symptoms when compared to those scabies participants recruited from the dermatology clinic. Skin NTDs like Buruli ulcer have been associated with supernatural causes by persons in endemic communities [29]. In another study in Ghana, 13% of participants believed that yaws was caused by supernatural forces such as witchcraft, curses or a punishment from god and this belief was not associated with gender, religion or ethnicity [24]. The similarity of cultural, religious and ethnic backgrounds may account for the observed belief systems regarding scabies in the current study. Such beliefs in supernatural forces as the cause of scabies may further delay health seeking and the initiation and completion of appropriate treatment for the disease in endemic communities. Witchcraft and curses are more often linked to diseases which are difficult to cure, take long time to heal and those whose causation are difficult to explain. Scabies however is a rather acute disease that is easy to treat (if treatment is available). A quicker diagnosis and effective treatment of scabies can lead to people less frequently associating scabies with witchcraft and/ or curses.

Beliefs, attitudes and experiences can shape health-seeking behaviour in scabies endemic communities. Delayed care seeking in scabies can promote on-going community transmission and outbreaks [5]. We found that a perceived lack of disease severity was the most common reason for delay in health seeking among scabies participants. This is similar to findings in studies on Buruli ulcer [30, 31] and leprosy [32]. In a study of yaws and yaws like skin diseases in Ghana, cost of treatment was reported as the most important factor in determining the choice and place of care. While 92.9% of affected individuals indicated that direct cost of treatment was important, 55.6% indicated that the cost of travel was an important consideration in the choice of care. The opinions of family members or an individual's previous experience were relatively infrequent considerations in guiding treatment seeking behaviour [24]. In the present study, cost of treatment was mentioned by 26.9% of participants with scabies but only 9% mentioned cost of travel as an important consideration for health seeking choice. Other important drivers of delayed care seeking were use of alternative treatments (13.4%) and a perceived ineffectiveness of the treatment provided by health workers (10.4%). It is important to note that although cost of treatment was the second leading reason for delay in care seeking, treatment for scabies is free under the national health insurance scheme in Ghana. Educational programmes should therefore emphasise availability of free and effective treatment for persons with health insurance. Since scabies often occurs in outbreaks, it is more difficult to predict how many scabies treatments are needed; therefore there is a risk of treatment not being available at the health facility level. Non-availability of drugs at the facility level to treat cases and their contacts will reflect poorly and result in low confidence in the health system. Health systems should be capacitated to readily scale up to deal with scabies outbreaks. Distance did not appear to be an important factor in delayed care seeking and was mentioned by only 4.5% of respondents. Indeed, in a previous study on scabies in rural Ghana, 48.4% of patients bypassed their nearest clinic, travelling a mean 6.2 km more than they theoretically needed to in order to access care [33]. In the current study, more participants recruited from the dermatology clinic cited possession of health insurance as a reason for seeking care in the formal health system. Despite this, there is need to enhance access to care and improve drug availability (on

health insurance) in a responsive way. This has the potential to enhance trust in the formal health system, which together with shared positive experiences of persons previously treated by health workers were found to be important factors in the decision making. Scabies control should be integrated in the activities of other neglected tropical disease programmes in Ghana in line with current recommendations [34]. Additionally, the deployment of a decentralised model of care as advocated for skin NTDs like Buruli ulcer [35–37] has the potential to make care accessible and affordable for affected persons and help address scabies burden in poor and marginalised communities which face the greatest burden of the disease.

Oral ivermectin has been shown to be effective for mass drug administrations (MDA) for scabies and current recommendations by the WHO and international scabies organisations [26, 38] propose its use for dealing with scabies outbreaks. For low resource settings like Ghana, there is a need to improve recognition, diagnosis and reporting of scabies to adequately document the disease burden in order to garner community support for scabies control efforts.

The perceived ineffectiveness of treatment provided by health workers for scabies was a contributor to delayed care seeking by a few participants. In a study of community members in Buruli ulcer endemic regions of Benin, an individual's perception on the effectiveness of treatment and the timeline of disease were the factors that most influenced pre-hospital delay by healthy individuals [30].

Aside delaying care seeking, supernatural beliefs on causation can result in blaming and stigmatizing of affected persons. We found that stigma was present in scabies. Experiences of felt stigma in scabies were reported as avoidance and shaming; similar findings have been observed in Buruli ulcer patients [22]. Studies from Brazil [39] and Guinea Bissau [11] have reported scabies to be associated with experiences such as shaming, social restrictions, behavioural changes and stigmatization. Furthermore, stigma may result in delayed health-seeking or a withdrawal from treatment, but this can be influenced positively or negatively by families and health care workers. There is the need to increase community awareness of negative effect of such factors on affected individuals to facilitate control activities.

In this study, participants with scabies reported the disease caused itchiness which impacted their sleep and productivity at work. The median age for individuals with scabies in this study was 29 years; impaired productivity resulting from scabies can have socio-economic consequences for the affected individuals and their families. In a recent study from the Solomon Islands, scabies was found to have a measurable impact on health-related quality of life with the greatest impact due to itching, sleep disturbances and impacts on education and employment [40]. There is a need to conduct further studies to fully ascertain the health and socio-economic impact of scabies in Ghana including how it affects schooling.

Limitations of study

The diagnosis of scabies by community members who reported having scabies in the past year may have been inaccurate. However, since the aim of the study was to assess ideas on scabies, we included participants who thought they had scabies and not only persons with diagnostically confirmed scabies. There might have been a recall bias by participants regarding a past diagnosis of scabies. We sought to limit this by including only individuals with a past diagnosis <1 year ago as we believe that information about a more recent diagnosis is more reliable than that about a diagnosis made long ago. A good proof about a history of scabies may be obtained from the medical records of participants. However, we were unable to access the medical records of participants with past scabies recruited from the community setting. It was only possible to refer to the medical records of those scabies participants recruited from the dermatology clinic. Interviews were done by health workers so there is a risk that respondents may have provided socially acceptable answers. Further, the study was done in the Ashanti

region where most persons are of Akan ethnicity so the findings may not be representative of the belief systems of other ethnic groups in Ghana. However, the Akan ethnic group is the largest in the country and make up 47.5% of the population [41]. Despite the limitations, this is the first study that has assessed beliefs, attitudes and practices towards scabies in Ghana. Our findings provide important information to guide the development of community control programmes for scabies and research on effect of education on health seeking as part of integrated control for neglected tropical diseases in Ghana.

## Conclusion

Although actual known causes of scabies were quite often mentioned by participants, there were several misconceptions about scabies. A belief in supernatural cause of scabies was lower in scabies participants recruited from the clinic setting; this finding may indicate that the belief system influences where affected individuals go to seek care. Delayed care seeking for the disease is due to several factors including a perceived lack of disease severity, treatment cost and use of alternative remedies and perceived ineffectiveness of treatment. A quicker diagnosis and effective treatment of scabies can lead to community members less frequently associating scabies with witchcraft and/ or curses. There is the need to enhance health education to promote early care seeking, enhance knowledge of communities on impact and dispel negative perceptions about scabies in Ghana.

## Acknowledgments

We thank the Informatiemanagement Onderzoek at the University Medical Centre Groningen for the greatly appreciated support with REDCap mobile app. We are grateful to Abigail Agbanyo and Bernadette Agbavor of the Kumasi Centre for Collaborative Research for their assistance.

## Author Contributions

**Conceptualization:** Yaw Ampem Amoako, Richard Odame Phillips, Ymkje Stienstra.

**Data curation:** Yaw Ampem Amoako, Lotte Suzanne van Rietschoten, Michael Ntiamoah Oppong, Kwabena Oppong Amoako, Kabiru Mohammed Abass, Dennis Odai Laryea.

**Formal analysis:** Yaw Ampem Amoako, Dennis Odai Laryea.

**Investigation:** Yaw Ampem Amoako, Lotte Suzanne van Rietschoten, Michael Ntiamoah Oppong, Kwabena Oppong Amoako, Kabiru Mohammed Abass, Bernard Akoto Anim.

**Methodology:** Yaw Ampem Amoako, Richard Odame Phillips, Ymkje Stienstra.

**Project administration:** Yaw Ampem Amoako, Michael Ntiamoah Oppong, Richard Odame Phillips, Ymkje Stienstra.

**Supervision:** Richard Odame Phillips, Ymkje Stienstra.

**Validation:** Ymkje Stienstra.

**Visualization:** Bernard Akoto Anim, Dennis Odai Laryea.

**Writing – original draft:** Yaw Ampem Amoako.

**Writing – review & editing:** Yaw Ampem Amoako, Lotte Suzanne van Rietschoten, Michael Ntiamoah Oppong, Kwabena Oppong Amoako, Kabiru Mohammed Abass, Bernard Akoto Anim, Dennis Odai Laryea, Richard Odame Phillips, Ymkje Stienstra.

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
