## [Decision Letter · Decision Letter 0]

4 Jan 2023

Dear Dr. Amoako,

Thank you very much for submitting your manuscript "Beliefs, attitudes and practices towards scabies in central Ghana" for consideration at PLOS Neglected Tropical Diseases. As with all papers reviewed by the journal, your manuscript was reviewed by members of the editorial board and by several independent reviewers. In light of the reviews (below this email), we would like to invite the resubmission of a significantly-revised version that takes into account the reviewers' comments. 

The reviewers felt that this manuscript is of general public health importance as it is the first study to attempt to assess beliefs, attitudes and practices towards scabies in Ghana. The paper is also of interest as it indicated that some people seemed unaware that treatment for scabies is free under the national health insurance scheme in Ghana which highlights the need for educational programmes to promote the availability of free and effective treatment for persons with health insurance. There are some significant concerns, however, regarding the methodology and conclusions from the results. These are outlined in detail by reviewer number 2 and each of these points need to be addressed prior to any re-submission. 

We cannot make any decision about publication until we have seen the revised manuscript and your response to the reviewers' comments. Your revised manuscript is also likely to be sent to reviewers for further evaluation.

Sincerely,

Michele Murdoch

Guest Editor

Joseph Vinetz

Section Editor

Thank you to the authors for submitting this manuscript, which is of general public health importance as it is the first study to attempt to assess beliefs, attitudes and practices towards scabies in Ghana. The paper is also of interest as it indicated that some people seemed unaware that treatment for scabies is free under the national health insurance scheme in Ghana which highlights the need for educational programmes to promote the availability of free and effective treatment for persons with health insurance. There are some significant concerns, however, regarding the methodology and conclusions from the results. These are outlined in detail by reviewer number 2 and each of these points need to be addressed prior to any re-submission. 

Additional minor comment:

Fig 4. Does this reflect the data from only the scabies group? If so, please amend the legend title accordingly.

Reviewer's Responses to Questions

**Key Review Criteria Required for Acceptance?**

**Methods**

-Are the objectives of the study clearly articulated with a clear testable hypothesis stated?

-Is the study design appropriate to address the stated objectives?

-Is the population clearly described and appropriate for the hypothesis being tested?

-Is the sample size sufficient to ensure adequate power to address the hypothesis being tested?

-Were correct statistical analysis used to support conclusions?

-Are there concerns about ethical or regulatory requirements being met?

Reviewer #1: an excellent and relevant study which was well conducted

Reviewer #2: Not clear how the sample size was reached and how the actual recruitments took place

Reviewer #3: The objective of the study is clearly articulated though there is no hypothesis stated. The study design is appropriate for the objective. There are no ethical concerns regarding this study.

**Results**

-Does the analysis presented match the analysis plan?

-Are the results clearly and completely presented?

-Are the figures (Tables, Images) of sufficient quality for clarity?

Reviewer #1: well analysed and presented

Reviewer #2: No comment

Reviewer #3: The analysis presented corresponds to the analysis plan and the results are clearly and completely presented and the Figures are clear.

**Conclusions**

-Are the conclusions supported by the data presented?

-Are the limitations of analysis clearly described?

-Do the authors discuss how these data can be helpful to advance our understanding of the topic under study?

-Is public health relevance addressed?

Reviewer #1: Conclusions are supported and there are recommendations stemming from this work

Reviewer #2: I do not think the conclusions are fair reflection of the findings. Details in the attachment

Reviewer #3: The conclusions support data represented and the limitations are mentioned. The data and its usefulness understanding the topic are discussed as well as its public health relevance in development of community control programs and recommendation for enhancement of health education in Ghana and availability of scabies treatment by the Government

**Editorial and Data Presentation Modifications?**

Reviewer #1: Accept

Reviewer #2: (No Response)

Reviewer #3: Overall, the paper is well written and there are few grammatical edits to be made in addition to the details of the clinical diagnostic criteria of scabies in this study.

**Summary and General Comments**

Reviewer #1: An important study on a common and distressing condition in communities. Understanding underlying belief systems and providing insights into the cause is key to encouraging earlier presentation and improving the quality of life of those affected

Reviewer #2: Possible bias in the interpretation of the result where the authors have a message they want to convey and totally disregard the data at their disposal but follow faint leads to their conclusions

Reviewer #3: This paper reports on the Beliefs, attitudes, and practices towards scabies in central Ghana. It presents important findings that are relevant in understanding the problem in the community and how to tackle them to ensure scabies control. The discussion is relevant and extensive and covers the topic appropriately.

PLOS authors have the option to publish the peer review history of their article (what does this mean?). If published, this will include your full peer review and any attached files.

Reviewer #1: Yes: Anisa Mosam

Reviewer #2: No

Reviewer #3: No
---

## [Decision Letter · Decision Letter 1]

13 Feb 2023

Dear Dr. Yaw,

We are pleased to inform you that your manuscript 'Beliefs, attitudes and practices towards scabies in central Ghana' has been provisionally accepted for publication in PLOS Neglected Tropical Diseases.

Best regards,

Michele Murdoch

Guest Editor

Joseph Vinetz

Section Editor

Reviewer's Responses to Questions

**Key Review Criteria Required for Acceptance?**

**Methods**

-Are the objectives of the study clearly articulated with a clear testable hypothesis stated?

-Is the study design appropriate to address the stated objectives?

-Is the population clearly described and appropriate for the hypothesis being tested?

-Is the sample size sufficient to ensure adequate power to address the hypothesis being tested?

-Were correct statistical analysis used to support conclusions?

-Are there concerns about ethical or regulatory requirements being met?

Reviewer #1: This is a revised version which is much improved after all the reviewers queries were satisfactorily answered.

Reviewer #2: This is well outlined in the article

**Results**

-Does the analysis presented match the analysis plan?

-Are the results clearly and completely presented?

-Are the figures (Tables, Images) of sufficient quality for clarity?

Reviewer #1: This is a revised version which is much improved after all the reviewers queries were satisfactorily answered.

Reviewer #2: Yes

**Conclusions**

-Are the conclusions supported by the data presented?

-Are the limitations of analysis clearly described?

-Do the authors discuss how these data can be helpful to advance our understanding of the topic under study?

-Is public health relevance addressed?

Reviewer #1: This is a revised version which is much improved after all the reviewers queries were satisfactorily answered.

Reviewer #2: Yes

**Editorial and Data Presentation Modifications?**

Reviewer #1: Accept

Reviewer #2: Accept

**Summary and General Comments**

Reviewer #1: (No Response)

Reviewer #2: Accept

PLOS authors have the option to publish the peer review history of their article (what does this mean?). If published, this will include your full peer review and any attached files.

Reviewer #1: **Yes: **Anisa Mosam

Reviewer #2: No

---

## [Editor Report · Acceptance letter]

20 Feb 2023

Dear Amoako,

We are delighted to inform you that your manuscript, "Beliefs, attitudes and practices towards scabies in central Ghana," has been formally accepted for publication in PLOS Neglected Tropical Diseases.

Best regards,

Shaden Kamhawi

co-Editor-in-Chief

Paul Brindley

co-Editor-in-Chief
